# The effect of rock composition on muon tomography measurements

Alessandro Lechmann[1], David Mair[1], Akitaka Ariga[2], Tomoko Ariga[3], Antonio Ereditato[2], Ryuichi Nishiyama[2], Ciro Pistillo[2], Paola Scampoli[2,4], Fritz Schlunegger[1], Mykhailo Vladymyrov[2]

[1]Institute of Geological Sciences, University of Bern, Bern, CH-3012, Switzerland
[2]Albert Einstein Center for Fundamental Physics, Laboratory for High Energy Physics, University of Bern, Bern, CH-3012, Switzerland
[3]Faculty of Arts and Science, Kyushu University, Fukuoka, J-812-8582, Japan
[4]Dipartimento di Fisica "E.Pancini", Università di Napoli Federico II, Napoli, I-80126, Italy

*Correspondence to*: Alessandro Lechmann (alessandro.lechmann@geo.unibe.ch)

**Abstract.** In recent years, the use of radiographic inspection with cosmic-ray muons has spread into multiple research and industrial fields. This technique is based on the high-penetration power of cosmogenic muons. Specifically, it allows the resolution of internal density structures of large scale, geological objects through precise measurements of the muon absorption rate. So far, in many previous works, this muon absorption rate has been considered to depend solely on the density of traversed material (under the assumption of a standard rock) but the variation in chemical composition has not been taken seriously into account. However, from our experience with muon tomography in Alpine environments we find that this assumption causes a substantial bias on the muon flux calculation, particularly where the target consists of high $\{Z^2/A\}$ rocks (like basalts and limestones) and where the material thickness exceeds 300 metres. In this paper, we derive an energy loss equation for different minerals and we additionally derive a related equation for mineral assemblages that can be used for any rock type on which mineralogical data are available. Thus, for muon tomography experiments in which high $\{Z^2/A\}$ rock thicknesses can be expected, it is advisable to plan an accompanying geological field campaign to determine a realistic rock model.

## 1 Introduction

The discovery of the muon (Neddermeyer and Anderson, 1937) entailed experiments to characterise its propagation through different materials. The fact that muons lose energy proportionally to the mass density of the traversed matter (see Olive et al., 2014) inspired the idea of using their attenuation to retrieve information on the traversed material. This was first done by George (1955) for the estimation of the overburden upon building of a tunnel, and then later by Alvarez et al. (1970) to search for hidden chambers in the pyramids at Giza (Egypt). In a related study, Fujii et al. (2013) employed this technology to locate the reactor of a nuclear power plant. Recently, Morishima et al. (2017) successfully accomplished quest of Alvarez' team in the Egyptian Pyramids.

Besides these applications, which have mainly been designed for archaeological and civil engineering purposes, scientists have begun to deploy particle detectors to investigate and map geological structures. In recent years this has been done for various volcanoes in Japan (Nishiyama et al., 2014; Tanaka et al., 2005, 2014), including Shinmoe-dake volcano (Kusagaya and Tanaka, 2015), the lava dome at Unzen (Tanaka, 2016) and most recently Sakurajima volcano (Oláh et al., 2018).

Further experiments have been conducted in the Caribbean, in France (Ambrosino et al., 2015; Jourde et al., 2013, 2015; Lesparre et al., 2012; Marteau et al., 2015) and in Italy on Etna (Lo Presti et al., 2018) and Stromboli (Tioukov et al., 2017). Recently, Barnaföldi et al. (2012) used this technology to examine karstic caves in the Hungarian mountains. Our group is presently carrying out an experimental campaign in the Central Swiss Alps for the purpose of imagining glacier-bedrock interfaces (Nishiyama et al., 2017).


Inferences about subsurface structures from observed muon flux (i.e. the number of recorded muons normalised by the exposure time and the detector acceptance) necessitate a comparison of the measurement data with muon flux simulations for structures with various densities. Such a simulation consists of a cosmic-ray muon energy spectrum model and a subsequent transportation of these muons through matter. The former describes the abundance of cosmic-ray muons for

different energies and zenith angles at the surface of the earth. This has been well documented in literature (see for example Lesparre et al., 2010). The differences between models and experimental data, hence the systematic model uncertainty, can be as large as 15 % for vertical muons (Hebbeker and Timmermans, 2002). On the other hand, the attenuation of the muon flux is assumed to depend only on the density of the traversed material. In this context, however, potential effects of its chemical composition have not been taken into account specifically. Instead, previous works employ a certain representative

rock, so-called "standard rock", for which the rate of muon energy loss has been tabulated (e.g. Groom et al., 2001).

The origin of this peculiar rock type can be traced back to Hayman et al. (1963), Miyake et al. (1964), Mandò and Ronchi (1952) and George (1952), who gave slightly different definitions of its physical parameters (mass density $\rho$, atomic weight $A$ and atomic number $Z$). A comprehensive compilation thereof can be found in Table 1 of Higashi et al. (1966). Various

corrections to the energy loss equation were then added in the framework of following-up studies, which particularly includes a density effect correction (see for example Sternheimer et al., 1984). Richard-Serre (1971) listed data relevant for muon attenuation for: (i) soil from the CERN (European Organization for Nuclear Research) premises near Geneva (Switzerland), (ii) Molasse-type material (e.g. Matter et al., 1980) and (iii) a "rock" that equals the one from Hayman et al. (1963). These latter authors assigned additional energy loss parameters to this particular rock type, which were similar to

those of pure quartz. Lohmann et al. (1985) then adjusted these parameters to energy loss variables for calcium carbonate (i.e. calcite) and gave the standard rock its present shape. In summary, this fictitious material consists of a density of crystalline quartz (i.e. $\rho_{qtz} = 2.65 \ g \ cm^{-3}$), a Z and A of 11 and 22, respectively (which is almost sodium), and density effect parameters that have been measured on calcium carbonate.

However, when the material's Z and A differ greatly from standard rock parameters as for carbonates, basalts or peridotites, a substantial bias would be introduced to the calculation of the muon flux. Such a situation is easily encountered in geological settings such as the European Alps where igneous intrusions, thrusted and folded sedimentary covers and recent Quaternary deposits are found in close vicinity (e.g. Schmid et al., 1996). Currently, our collaboration is performing a muon tomography experiment in the Jungfrau region, in the Central Swiss Alps aiming at imaging the glacier-bedrock interface

(Ariga et al., 2018; Nishiyama et al., 2017). There, we face a variety of lithologies ranging from gneissic to carbonatic rocks that have a thickness larger than 500 m (Mair et al., 2018). In this context, it turned out that the analyses based on the standard rock assumption might cause an over- or an underestimation of the bedrock position in the related experiment. Such an uncertainty arising from the chemical composition of the actual rock has to be reduced at least to the level of the statistical uncertainty inherent in the measurement as well as in the systematic uncertainty of the muon energy spectrum model.

To achieve this, we investigate how different rock types potentially influence the results of a muon tomographic experiment. We particularly compare the lithologic effect on simulated data with standard rock data to estimate a systematic error that is solely induced by a too simplistic assumption on the composition of the bedrock.

## 2 Methods

### 2.1 Rock types

In this study, we chose 10 different rock types that cover the largest range of natural lithologies, spanning the entire range from igneous to sedimentary rocks. The simplest rocks have a massive fabric in the sense that they do not exhibit any planar or porphyric texture. Typical lithologies with these characteristics are igneous rocks or massive limestones (not sandstones as they might have a planar fabric such as laminations and ripples). Exemplary thin sections of a granite and a limestone are shown in Fig. 1. Note that rocks featuring strong heterogenic, metamorphic textures are not treated in the framework of this

study for simplicity purposes and will be subject of future research. Also, for simplicity purposes, we do not consider spatial variations in crystal sizes in our calculations (i.e. a porphyric texture). We justify this approach because a related inhomogeneity is likely to be averaged out if one considers a several metre-thick rock column. Additionally, the rock is considered to consist only of crystalline components, i.e. glassy materials such as obsidian have to be treated separately. Porous media can be approximated by assigning one of the constituents as air or (in the case of a pore fluid) water. This is

explicitly done for the case of arkoses (10% air) and arenites (11% air).

We compare the energy loss of muons in these rocks and hence the resultant muon flux attenuation depending on depth with those of the standard rock. The analysed lithologies, together with their relevant physical parameters, are listed in Table 1. Among these parameters, $\{Z/A\}$ and $\{Z^2/A\}$, i.e. the ratio of the atomic number (and its square) to the mass number

averaged over the entire rock, are most relevant to the energy loss of muons (Groom et al., 2001). The former is almost proportional to the ionisation energy loss that occurs predominantly at low energies, whereas the latter is mostly proportional

to the radiation energy loss, that becomes dominant for muons faster than their critical energy at around $600\ GeV$. The volumetric mineral fractions of these ten rocks can be found in Appendix A.

## 2.2 Cosmic ray flux model

We perform our calculations with the muon energy spectrum model proposed by Reyna (2006), at sea level and for vertical incident muons. This model describes the kinetic energy distribution of the muons before they enter the rock. The calculation of the integrated muon flux after having crossed a certain amount of material is done in two steps. First, the minimum energy required for muons to penetrate a given thickness of rock is calculated considering the chemical composition effects (see Sect. 2.3). Afterwards, the energy spectrum model, $dF/dE$, is integrated above the obtained minimum energy (which we call from here on "cut-off energy", $E_{cut}$) to infinity, i.e.

$$F_{calc} = \int_{E_{cut}}^{\infty} \frac{dF(E)}{dE}\ dE. \tag{1}$$

The integration is necessary as most detectors, which have been used for muon tomography, record only the integrated muon flux. As already stated in the introduction, we attribute a systematic uncertainty of $\pm$ 15 % to the integrand $dF/dE$. All the calculations in this work have been verified with another flux model (Tang et al., 2006) and are presented in the supplementary material.

## 2.3 Muon propagation in rocks

As soon as muons penetrate a material, they start to interact with the material's electrons and nuclei and lose part of their kinetic energy. The occurring processes can be categorised into an ionisation process, i.e. a continuous interaction with the material's electrons, and radiative interactions with the material's nuclei (i.e. bremsstrahlung, electron-positron pair production and photonuclear processes), which are of a stochastic nature. All these processes are governed by the material density $\rho$ and the atomic number Z and atomic weight A (see Groom et al., 2001 for details). Our general strategy for the calculation of the energy loss in a rock is to use its decomposition into energy losses for the corresponding minerals. Accordingly, the energy loss of muons travelling a unit length, $dE/dx$, in a rock can be described by a volumetrically averaged energy loss through its mineral constituents

$$\left\{ \frac{dE_{rock}}{dx} \right\} = \sum_j \varphi_j \left\langle \frac{dE_{mineral,j}}{dx} \right\rangle, \tag{2}$$

where $\varphi_j$ is the volumetric fraction of the j-th mineral within the rock. The derivation of Eq. (2) can be found in Appendix B. In order to exploit this abstraction efficiently we have to assume a homogeneous mineral distribution within the rock. This is a strong simplification, considering for example effects related to a local intrusion, tectonic processes like folding and thrusting, or spatial differences in sedimentation patterns. These concerns can be addressed through averaging over a large enough volume. Figure 1 shows two typical thin-sections from rock samples of our experimental site that exhibit crystal

sizes well below 4 mm - 5 mm. As muon tomography for geological purposes generally operates at scales of 10 m - 1000 m it is safe to assume that small-scale variations are averaged out. Thus, the term on the right-hand side of Eq. (2), i.e. the energy loss across each mineral, can be written as:

$$- \left\langle \frac{dE_{mineral}}{dx} \right\rangle = \rho_{mineral} * (\langle a \rangle + E * \langle b \rangle), \tag{3}$$

where $\langle a \rangle$ and $\langle b \rangle$ are the ionisation and radiative energy losses across a given mineral, respectively. These two parameters are in turn calculated by averaging the contribution of each element (i.e. atom) constituting the mineral by their mass (see Eq. (B5) to Eq. (B15) in Appendix B for details). The density of the minerals, $\rho_{mineral}$, is estimated from its crystal structures (see Appendix A for more detailed instructions). Once the energy losses are obtained for all minerals, each contribution is summed up according to Eq. (2). The energy loss within the rock can then be expressed in a similar way, as in

Eq. (3), (for a detailed discussion we refer to Appendix B):

$$- \left\{ \frac{dE_{rock}}{dx} \right\} = \rho_{rock} * (\{a\} + E * \{b\}). \tag{4}$$

Again, the values $\{a\}$ and $\{b\}$ indicate the averaged ionisation and radiative energy losses across the whole rock, respectively. Equation (4), an ordinary nonlinear differential equation, is usually given as a final value problem, i.e. we know that the muon, after having passed through the rock column, still needs some energy to penetrate the detector, $E_{det}$. This can

be turned into an initial value problem, by reversing the sign of Eq. (4) and defining the detector energy threshold as initial condition.

$$\left\{ \frac{dE_{rock}}{dx} \right\} = \rho_{rock} * (\{a\} + E * \{b\}) \tag{5}$$

$$E(x = 0) = E_{det}$$

The problem has been transformed into the one of finding the final energy, the cut-off energy, $E_{cut}$, after a predefined

thickness of rock. This is a well investigated problem, for which a great variety of numerical solvers are available. In this work we employ a standard Runge-Kutta integration scheme (see for example Stoer and Bulirsh, 2002).

The energy loss equations are subject to systematic uncertainties, mainly because the experimentally determined interaction cross sections have an attributed error. According to Groom et al., (2001), the error on ionisation losses is "mostly smaller than 1 % and hardly ever greater than 2 %". These authors also state, that in the case of compounds the uncertainties might

be thrice as large. Therefore, we considered an ionisation loss uncertainty of $\pm$ 6 % as appropriate for our calculations. The errors on the cross sections of bremsstrahlung, pair-production and photonuclear interactions are $\pm$ 1 %, $\pm$ 5 % and $\pm$ 30 %, respectively. Appendix C shows in detail how we propagated these errors to the cut-off energy, $E_{cut}$.

## 3 Results

Figures 2 and 3 show the muon flux simulations as a function of rock thicknesses up to 2 km for igneous and sedimentary

rocks, respectively. The depth-intensity relation is described by a power law, as it is the integration of the differential energy spectrum of muons, which also follows a power law.

To better visualise the difference between the fluxes after having passed these ten rock types and the standard rock, we report the ratio between fluxes calculated after the different materials and that after the standard rock in Fig. 4:

$$fr_{\text{rock}} = \frac{F_{calc,rock}}{F_{calc,SR}} .$$  (6)

The attenuation of the muon flux expectedly depends predominantly on the rock density, as we can see in Figs. 2 to 4. Rocks exhibiting a high material density result in a larger muon flux attenuation than lithologies with a lower density. This however, only depicts the overall differences, including density and compositional variations, between real and standard rock. In this regard, Groom et al. (2001) apply an explicit treatment of density variations of known materials. Thus, the flux data can be simulated for a standard rock with the exact density as its real counterpart. Such a density normalisation enables

us to isolate the compositional influence on the computed data. Figures 5 and 6 show the muon flux simulations for each rock compared to a density normalised standard rock and Fig. 7 summarises this information by representing the ratio between muon fluxes after passing through real rocks and the muon flux after passing through a density normalised standard rock. It is important to note that the standard rock muon flux in each flux ratio has been normalised with respect to the density of the original rock (i.e. the peridotite is compared to a standard rock of density $\rho = 3.340\ g\ cm^{-3}$, the limestone is

comparted to a standard rock of density $\rho = 2.711\ g\ cm^{-3}$, etc.). One notices that the flux ratios are rather close together, mainly within 2.5 % of the standard rock flux, before they start to diverge towards larger (dolomite, shale and arenite) and smaller (igneous rocks, arkose, limestone and aragonite) flux ratios beyond 300 m thickness of penetrated rock. Even though the errors on the fluxes are relatively large and sometimes even overlap with the standard rock fluxes, the propagated errors on the flux ratios remain well bounded near their means. This effect is due to the correlation of the errors in the numerator

and the denominator in Eq. (6). A detailed discussion of how uncertainties have been propagated is presented in Appendix C.

## 4 Discussion

The differences in the calculated muon flux illustrated in Figs. 2 and 3 become even more pronounced in Fig. 4, where the fluxes are compared to the case where cosmic fluxes are attenuated by a standard rock. One notices a direct correlation with

material density. This is reinforced by the fact that the granite (Fig. 2) has the same density as the standard rock, 2.650 $g\ cm^{-3}$, and shows an overall similar flux magnitude as the standard rock, i.e. a flux ratio of 1. This can be explained by Eq. (4), as the energy loss is almost directly proportional to the density, while the presence of density in the ionisation loss term

(i.e. $\{a(E, \rho, A, Z)\}$ ) is negligible compared to this factor. Thus, if the rock flux data are compared to a standard rock with equal density, this effect should be removed, and one is left with the composition difference only.


A closer look at Fig. 7 reveals that the muon fluxes for every rock below 300 m do not depart more than 2.5 % from their respective density modified standard rock flux. The chemical composition effect can thus be considered negligible when compared to the systematic uncertainty originating from the muon flux model. We explain this through the dominance of the ionisation energy loss in this thickness region. Muons that penetrate down to 300 m of rock are still slow enough to
predominantly lose their kinetic energy for the ionisation of the rock's electrons. As the number of electrons per unit volume is given by the product: $\rho_{rock} * \{Z/A\}$, ionisation losses are proportional to this term. When comparing a density normalised standard rock with a real rock, the only difference can emerge from the second part, i.e. $\{Z/A\}$. According to Table 1, these values do not change more than 1 % with respect to each other.

When the rock thicknesses become larger than 300 m, the flux ratios start to exceed $\pm$ 2.5 % and the ratio patterns diverge. This corresponds to the point where radiative losses start to become the dominant energy loss processes. The latter are interactions of the muon with the nuclei of the atoms within the rock and its cross section is mainly proportional to the square of the nucleus' charge (i.e. $\{Z^2/A\}$). Hence, rocks that exhibit a lower $\{Z^2/A\}$-value than a standard rock (e.g. dolomite, arenite and shale) attenuate the muon flux less (i.e. flux ratio > 1), while all igneous rocks as well as limestone,
aragonite and arkose, that have a higher $\{Z^2/A\}$-value attenuate the muon flux more, which results in a lower flux ratio.

The above results reflect only the most striking connections to the chemical composition of a rock. In reality however, the nature of muonic energy loss processes is much more complex than the shape of the flux ratios in Fig. 4 below 300 m suggests. The actual ionisation energy loss, Eq. (B27), is an interplay of the mean excitation energy $\{I\}$, i.e. the mean energy
needed to ionise a material's electrons, the material density $\rho_{rock}$ , $\{Z/A\}$ and various correction terms that depend on these parameters. These additional factors are also responsible for the non-linear behaviour of the flux ratios between 100 m and around 600 m, as effects from radiative losses start to become significant. However, as the resulting differences due to these processes remain smaller than 2.5%, a detailed discussion of these matters falls beyond the scope of this paper.

As we see above, the muon flux calculation is significantly biased when one employs the standard rock assumption and thus neglects the effect of the chemical composition, especially when the thickness of the rock is beyond 300 m. This systematic error would then later turn into an over- or an underestimation in the assessment of density structures. We can roughly estimate the error on a thickness estimation of a certain structure, by employing the following formula

$$\varepsilon_d(x_{\mathrm{ro}}(\mathrm{F})) = \frac{x_{SR}(F) - x_{Ro}(F)}{x_{Ro}(F)} . \tag{7}$$

Here, $x_{SR}(F)$ and $x_{Ro}(F)$ denote the thickness of standard rock and a real rock respectively, needed to attenuate the cosmic ray muon flux to $F$. This is possible because the flux, as a function of rock thickness, is a strictly decreasing function. The domain of this function ranges from zero to infinite thickness, where its image takes the values from the initial flux, $F_0$, to zero. On these two sets the function is a bijection and therefore an inverse function, $x(F)$, exists. Although its functional form might be unknown, it is still possible to interpolate between the simulated points. For our rocks, this is shown in Fig. 8.

As an example, in case where the target is 600 m thick and made of limestone ($\rho = 2.711\ g\ cm^{-3}$), the standard rock assumption underestimates the flux by 7 % – 8 % and thus overestimates the thickness by around 15 m or 2.5 %. The same is valid for basalt and aragonite.

The above discussion concentrates on calculations of the mean values of model parameters. A full description encloses also
the propagation of their uncertainties. The rather large error bounds on single flux calculations stems from the uncertainties in the flux model and in the interaction cross-sections. However, by taking a ratio, i.e. Eq. (6), of quantities with correlated errors, the resulting uncertainty on the ratio tends to cancel out. If the errors were propagated by linear operations, they would even cancel out perfectly. The small error-bars which are still present in Figs. 4, 7 and 8 can be seen as effects of the nonlinearity in the differential equation, Eq. (5).


Because this is a pure sensitivity study, we cannot offer distinct measurements to verify our predictions. The reason for this is mainly because dedicated experimental campaigns have not yet been conducted and thus such data are not available. We suggest that future studies in this field will address the composition issue and try to experimentally constrain this theoretical model. Nevertheless, our inferences are based on the same conceptual framework that has already been used for other
materials, including standard rock. As a result of this, we find significant differences if the rock parameters are changed, especially for rock thicknesses larger than 300 m.

## 5 Conclusions

Our results suggest that it is safe to use the standard rock approximation for all rock types up to thicknesses of ~300 m, as the flux ratio will mainly remain within 2.5 % of the standard rock flux, which generally lies within the cosmic ray flux model
error. However, we also find that beyond these thicknesses the use of the standard rock approximation and its density-modified version could lead to a serious bias. This mainly concerns basaltic and carbonate rocks. The flux error for these rock types increases with growing material thickness. It can be extrapolated, that the errors grow even further beyond 600 m of material thickness up to a point where any inference based upon this approximation becomes difficult. This is, however, a thickness range where muon tomography becomes increasingly hard to perform, as lower fluxes have to be counterbalanced
by larger detectors and longer exposure times.

In order to account for the composition of rock, it is advisable to undertake a geological study of the region alongside the muon tomography measurements, especially when faced with basaltic rocks or carbonates, which includes at the least the analysis of local rock samples. Auxiliary data could comprise rock density measurements (i.e. He-pycnometer or buoyancy experiments), chemical composition, and mineralogical information (i.e. X-Ray diffractometry/fluorescence measurements) as well as microfabric analyses (i.e. mineral and fabric identification on thin sections). This additional information may help to constrain solutions of a subsequent inversion to a potentially smaller set. The use of additional information, such as spatial information in the form of a geological map or a 3D model of the geologic architecture, is strongly encouraged, because it might greatly improve the state of knowledge about the physical parameters that are to be unravelled.


**Appendix A**

To estimate the mineral density, we assume that it can be calculated by dividing the mass of the atoms within the crystal unit cell by the volume of the latter (see for example Borchart-Ott, 2009):

$$\rho_{mineral} = \frac{Q * M}{N_A * V_{Unit\ Cell}} \ . \tag{A1}$$

In this equation, M is the total molar mass of one mineral "formula unit", Q is the number of formula units per unit cell and
$V_{Unit\ Cell}$ is the volume of the unit cell. The latter is calculated by the volume formula of a parallelepiped:

$$V_{Unit\ Cell} = \left\| \vec{a} \cdot (\vec{b} \times \vec{c}) \right\| \ . \tag{A2}$$

Eq. (A2) can be rewritten as

$$V_{Unit\ Cell} = \|\vec{a}\| \|\vec{b}\| \|\vec{c}\| \sqrt{1 + 2\cos(\alpha)\cos(\beta)\cos(\gamma) - \cos^2(\alpha) - \cos^2(\beta) - \cos^2(\gamma)} \ . \tag{A3}$$

Here, $\vec{a}, \vec{b}, \vec{c}$ denote the unit cell vectors, their lengths, $\| \cdot \|$ is measured in Ångströms, i.e. $10^{-10}\ m$, whereas $\alpha, \beta, \gamma$ are the
internal angles between those vectors. These six parameters can be looked up for each mineral in the crystallographic literature (e.g. Strunz and Nickel, 2001).

The volumetric percentages of the minerals that constitute the 10 investigated rock types are shown in Table A1 and Table A2. They were chosen as a reasonable compromise from literature values (e.g. Best, 2003; Tuttle and Bowen, 1958; Folk,
270   1980).

## Appendix B

### Energy loss in elements

The average spatial differential energy loss can be written in a rather simple form (Barrett et al., 1952):

$$-\left(\frac{dE(\rho,A,Z)}{dx}\right) = \rho * (a(E,\rho,A,Z) + E * b(E,A,Z)). \tag{B1}$$

Here, $\rho, A, Z$ denote the mass density, atomic weight and atomic number of the penetrated material, while $E$ is the kinetic energy of the penetrating, charged particle and $x$ is the position coordinate. The function $a(E,\rho,A,Z)$ in Eq. (B1) is the differential ionisation energy loss that accounts for the ionisation of electrons of the penetrated material. In the case of incident muons (i.e. electric charge $q_\mu = -1\ C$ and mass $m_\mu = 105.7\ MeV/c^2$), the relationship expressed in Eq. (B1) takes

the form:

$$a(E,\rho,A,Z) = K\frac{Z}{A}\frac{1}{\beta^2}\left[\frac{1}{2}\ln\left(\frac{2m_e c^2 \beta^2 \gamma^2 Q_{max}(E)}{I(Z)^2}\right) - \beta^2 - \frac{\delta(\rho,Z,A)}{2} + \frac{1}{8}\frac{Q_{max}^2(E)}{(\gamma m_\mu c^2)^2}\right] + \Delta\left|\frac{dE}{dX}\right|(Z,A). \tag{B2}$$

In this equation, $\beta, \gamma$ are the relativistic factors and are, therefore, a function of the kinetic energy $E$. The constant $m_e$ denotes the mass of the electron and $c$ is the speed of light. $Q_{max}$ is the highest possible kinetic recoil energy of scattered electrons in the medium, while $K$ is a constant incorporating information about the electron density. The function $\delta(\rho,Z,A)$

is a correction factor, which considers the mechanisms where the material becomes polarised at higher muon energies, with the consequence that the energy loss is weaker (Sternheimer, 1952). The last term in Eq. (B2) is another correction factor, which considers bremsstrahlung from atomic electrons (not the incident muon, which would be the term in Eq. (3)) that also appears at higher muon energies. A more detailed explanation of this equation and its parameters can for example be found in Olive et al., 2014. In contrast to Eq. (B2), the function $b(E,A,Z)$ describes all the radiative processes that become

dominant at higher velocities (above $\sim 600\ GeV\ c^{-1}$ for muons). This term includes energy losses due to bremsstrahlung, electron-positron pair production as well as photonuclear interactions. These different contributions can be written independently from each other:

$$b(E,A,Z) = b_{brems}(E,A,Z) + b_{pair}(E,A,Z) + b_{photonucl}(E,A,Z). \tag{B3}$$

Each process in Eq. (B3) is computed by integrating its differential cross-section with respect to every possible amount of

transferred energy:

$$b_{process} = \frac{N_A}{A}\int_0^1 \nu\frac{d\sigma_{process}}{d\nu}d\nu. \tag{B4}$$

Here, $N_A$ is the Avogadro number and $\nu = \varepsilon/E$ the fractional energy loss (whereas $\varepsilon$ is the absolute energy loss) for this process. Specific cross-sections for bremsstrahlung (Kelner et al., 1995, 1997), photonuclear (Bezrukov and Bugaev, 1981)

and pair-production (Nikishov, 1978) energy losses are used by Groom et al. (2001) for the calculations of their tables. As this pair-production cross-section involves the calculation of many computationally extensive dilogarithms, an equivalent cross-section (Kelner, 1998; Kokoulin and Petrukhin, 1969, 1971), which is used in GEANT4 (Agostinelli et al., 2003) by default, is used in our study.

**Energy loss in minerals**

Since the above equations are valid for pure elements, adjustments are needed for compounds (e.g. minerals) and mixtures thereof (e.g. rocks). Generally, it is advised to use the physical parameters for materials that have already been measured (see Seltzer and Berger, 1982 for a compilation). However, except for calcium carbonate (i.e., calcite) and silicon dioxide (i.e. quartz), no other minerals have been investigated. This also means that there is no standard approach available for considering natural rocks. Fortunately, for such materials a theoretical framework has been proposed (see for example Appendix A of Groom et al., 2001). The basic idea is to consider the compound as a single "weighted average"-material and the energy loss therein as a mass weighted average of its constituents' energy loss:

$$\langle \frac{dE_{mineral}}{d\chi} \rangle = \sum_i w_i \left( \frac{dE_{element,i}}{d\chi} \right). \tag{B5}$$

The weights $w_i$ are calculated according to the atomic weights $A_i$ of the elements

$$w_i = \frac{n_i A_i}{\sum_k n_k A_k} = \frac{m_{element,i}}{m_{mineral}}, \tag{B6}$$

and can then be used to calculate an average $\langle Z/A \rangle$ value

$$\langle \frac{Z}{A} \rangle = \sum_i w_i \frac{Z_i}{A_i}. \tag{B7}$$

Equivalently, the average $\langle Z^2/A \rangle$ value can be calculated according to

$$\langle \frac{Z^2}{A} \rangle = \sum_i w_i \frac{Z_i^2}{A_i}. \tag{B8}$$

One more change must be made to the ionisation loss Eq. (B2) in order to appropriately account for the change in the atomic structure that emerged due to chemical bonding of the elementary constituents. This is reflected in a modified mean excitation energy $\langle I \rangle$, which can be calculated by taking the exponential of the function

$$\ln\langle I \rangle = \frac{\sum_i w_i \frac{Z_i}{A_i} \ln(I_i)}{\sum_j w_j \frac{Z_j}{A_j}}, \tag{B9}$$

which is basically a weighted geometric average of the elementary mean excitation energies

$$\langle I \rangle = \sqrt[\Sigma_j w_j \frac{Z_j}{A_j}]{\prod_i I_i^{w_i \frac{Z_i}{A_i}}} . \tag{B10}$$

One has to pay attention that the mean excitation energies for some elements, $I_i$, can change quite significantly when they are part of a chemical bond. A guideline to address this issue can be found in Seltzer and Berger (1982). Equations (B7) to (B10) are still a consequence of Eq. (B5). However, there is one term in the function $\delta(\rho, Z/A)$ in Eq. (B2) that is calculated differently from Eq. (B5). This concerns the logarithm of the plasma energy of the compound, which for an element is given by (e.g. Olive et al., 2014):

$$\ln(\hbar\omega_p) = \ln\left(28.816 * \sqrt{\rho \frac{Z}{A}}\right). \tag{B11}$$

According to Eq. (B5) the plasma energy for a compound should be calculated the same way as the mean excitation energy in Eq. (B9). However, Sternheimer et al. (1982) and Fano (1963) explicitly advise us to use the arithmetic mean within the logarithm when dealing with an atomic mixture (i.e. a molecule), yielding

$$\ln\langle\hbar\omega_p\rangle = \ln\left(28.816 * \sqrt{\rho_{mineral} \left\langle\frac{Z}{A}\right\rangle}\right). \tag{B12}$$

This results in the modified ionisation energy loss:

$$\langle a(E, \rho_{mineral}, A, Z)\rangle = K \left\langle\frac{Z}{A}\right\rangle \frac{1}{\beta^2} \left[\frac{1}{2} \ln\left(\frac{2m_e c^2 \beta^2 \gamma^2 Q_{max}(E)}{\langle I(Z)\rangle^2}\right) - \beta^2 - \frac{\delta\left(\rho_{mineral}, \left\langle\frac{Z}{A}\right\rangle\right)}{2} + \frac{1}{8}\frac{Q_{max}^2(E)}{(\gamma m_\mu c^2)^2}\right] + \Delta\left|\frac{dE}{dX}\right|\left(\left\langle\frac{Z}{A}\right\rangle\right). \tag{B13}$$

The radiation loss for the compound, on the other hand, is only a linear combination of the radiation losses of its elementary constituents, Eq. (B3), yielding:

$$\langle b \rangle = \sum_i w_i b_i . \tag{B14}$$

The resulting Eq. (B15)

$$-\left\langle\frac{dE_{mineral}}{dx}\right\rangle = \rho_{mineral} * (\langle a \rangle + E * \langle b \rangle), \tag{B15}$$

has now the same form as the energy loss Eq. (B1) for elements and can be solved accordingly.

**Energy loss in rocks**

To obtain an energy loss equation for rocks, a similar procedure as for forming minerals through the assembly of elements
can be applied. Starting from Eq. (B5) we consider the energy loss for a rock as mass weighted average of the energy losses
of its mineral constituents

$$\langle\frac{dE_{rock}}{d\chi}\rangle = \sum_j q_j \langle\frac{dE_{mineral,j}}{d\chi}\rangle \,, \tag{B16}$$

where $q_j$ are the mass fractions of the j-th mineral within the rock, analogous to Eq. (B6),

$$q_j = \frac{n_j A_j}{\sum_l n_l A_l} = \frac{m_{mineral,j}}{m_{rock}} \,. \tag{B17}$$

Using $d\chi = \rho * dx$, Eq. (B16) then takes the following form:

$$\frac{1}{\rho_{rock}}\langle\frac{dE_{rock}}{dx}\rangle = \sum_j \frac{q_j}{\rho_{mineral,j}}\langle\frac{dE_{mineral,j}}{dx}\rangle \,. \tag{B18}$$

By inserting Eq. (B17) into Eq. (B18), one obtains

$$\frac{1}{\rho_{rock}}\langle\frac{dE_{rock}}{dx}\rangle = \frac{1}{m_{rock}}\sum_j \frac{m_{mineral,j}}{\rho_{mineral,j}}\langle\frac{dE_{mineral,j}}{dx}\rangle \,. \tag{B19}$$

Multiplying both sides with $\rho_{rock}$ and applying the definition of the density, $\rho = m/v$, that can also be written as $v = m/\rho$,
Eq. (B19) becomes

$$\langle\frac{dE_{rock}}{dx}\rangle = \frac{1}{v_{rock}}\sum_j v_{mineral,j}\langle\frac{dE_{mineral,j}}{dx}\rangle \,. \tag{B20}$$

If one sets $\varphi_j = v_{mineral,j}/v_{rock}$ , the volumetric fraction of the j-th mineral within the rock, Eq. (B20) transforms into the
compound equation for rocks

$$\langle\frac{dE_{rock}}{dx}\rangle = \sum_j \varphi_j \langle\frac{dE_{mineral,j}}{dx}\rangle. \tag{B21}$$

Analogue to the mineral case we can now define new average energy loss parameters for the rock, beginning with its overall
density

$$\rho_{rock} = \sum_j \varphi_j \rho_{mineral,j} \,. \tag{B22}$$

The average $\{Z/A\}$ is given by

$$\left\{\frac{Z}{A}\right\} = \sum_j \frac{\rho_{mineral,j}}{\rho_{rock}} \varphi_j \langle\frac{Z}{A}\rangle_j \tag{B23}$$

and similarly, the average $\{Z^2/A\}$ can be calculated according to

$$\left\{\frac{Z^2}{A}\right\} = \sum_j \frac{\rho_{mineral,j}}{\rho_{rock}} \varphi_j \left\langle\frac{Z^2}{A}\right\rangle_j \; . \tag{B24}$$

The rock's mean excitation energy is

$$\ln\{I\} = \frac{\sum_j \frac{\rho_{mineral,j}}{\rho_{rock}} \varphi_j \left\langle\frac{Z}{A}\right\rangle_j \ln\langle I\rangle_j}{\sum_l \frac{\rho_{mineral,l}}{\rho_{rock}} \varphi_l \left\langle\frac{Z}{A}\right\rangle_l} \; . \tag{B25}$$

The only difference between the rock calculation and the mineral calculation enters in the calculation of the plasma energy. While in the mineral case we were advised to use Eq. (B11) instead of what would naturally follow from the weighted average in Eq. (B5), we prefer to use the weighted average, Eq. (B21),

$$\ln\{\hbar\omega_p\} = \frac{\sum_j \frac{\rho_{mineral,j}}{\rho_{rock}} \varphi_j \left\langle\frac{Z}{A}\right\rangle_j \ln\langle\hbar\omega_p\rangle_j}{\sum_l \frac{\rho_{mineral,l}}{\rho_{rock}} \varphi_l \left\langle\frac{Z}{A}\right\rangle_l} \tag{B26}$$

for the case of rocks. The reason for this lies in the fact that the density effect operates on a nanometric scale, whereas
minerals, have generally sizes between several micrometres and a few centimetres. In the case of a mineral compound, the molecular structure comprises also a nanometric scale.

These parameters can then be rearranged into an ionisation loss term for a rock

$$\{a(E,\rho_{rock},A,Z)\} = K\left\{\frac{Z}{A}\right\}\frac{1}{\beta^2}\left[\frac{1}{2}\ln\left(\frac{2m_e c^2 \beta^2 \gamma^2 Q_{max}(E)}{\{I(Z)\}^2}\right) - \beta^2 - \frac{\delta\left(\rho_{rock},\left\{\frac{Z}{A}\right\}\right)}{2} + \frac{1}{8}\frac{Q_{max}^2(E)}{(\gamma m_\mu c^2)^2}\right] + \Delta\left|\frac{dE}{dX}\right|\left(\left\{\frac{Z}{A}\right\}\right) \; . \tag{B27}$$

Like Eq. (B14) the radiative losses can be rewritten as a weighted average of the mineral radiative losses

$$\{b\} = \sum_j \frac{\rho_{mineral,j}}{\rho_{rock}} \varphi_j \langle b\rangle_j \; . \tag{B28}$$

Equations. (B27) and (B28) can then be joined together to form again a similar term to Eqs. (B1) and (B15),

$$-\left\{\frac{dE_{rock}}{dx}\right\} = \rho_{rock} * (\{a\} + E * \{b\}) \; , \tag{B29}$$

the energy loss equation for rocks.

We want to stress that the starting point of the derivation of the energy loss equation for rocks is a mass averaging of mineral energy losses. Therefore, the mass averaging approach is inherently included in this approach. In fact, mass averaging and

volumetric averaging are two equivalent descriptions of the same problem. For the mass averaged formulae we refer to the supplementary material to this manuscript.

 **Appendix C**

**Uncertainty propagation**

The first step in our uncertainty treatment includes a propagation of the interaction cross section errors ( $\sigma_a = \pm 6\,\%, \sigma_{b_{brems}} = \pm 1\,\%, \sigma_{b_{pair}} = \pm 5\,\%, \sigma_{b_{photonucl}} = \pm 30\,\%$ ) to the cut-off energy, i.e. by solving the differential equation Eq. (5). Generally, a higher cross section yields a higher cut-off energy, as the muon needs more initial kinetic energy, which

it then loses on the way and vice-versa. In order to estimate a lower and an upper error bound on the cut-off energy, $E_{cut}$, we use a conservative approach. This means that the lower cut-off energy error bound is calculated by setting all cross sections to their lower $1\,\sigma$ bound and running the simulation with these modified values. The upper error bound is calculated accordingly. Of course, this overestimates the effective error, however if our calculations remain valid within this conservative error, then they can also be trusted with a conventional error.


The second step is the estimation of the error regarding the integrated flux. Here we need to propagate the errors through Eq. (1) to the simulated flux. There are two different errors present at this stage. The first one includes the error on the lower integration boundary, i.e. $E_{cut}$, which has just been calculated above. The second error addresses the integrand, i.e. the flux model. Figure C1 visualises the concept behind the propagation of these two errors. The simulated flux error is equivalent to

the error, which is made by calculating the area under the graphs. We estimate the lower error bound on the simulated flux (i.e. smallest area), by taking the upper error bound on $E_{cut}$ and the lower error bound on $dF/dE$. Similarly, the upper error bound on the simulated flux (i.e. largest area) is calculated by setting $E_{cut}$ to its lower error bound and $dF/dE$ to its upper error bound. Again, this is a conservative approach, which we justify with the same rationale as above.

The last step addresses the propagation of the simulated flux errors to the flux ratio in Eq. (6). Here we can make use of the fact that the errors in both simulations are perfectly correlated. In other words, if we knew the errors on all affected quantities in one simulation, we would instantaneously know the corresponding values for any other simulation. This allows us, for example, to calculate the upper error bound on the flux ratio by dividing the upper error bound of the simulated flux in the numerator by the upper error bound of the simulated flux in the denominator. The same is valid for any other

constellation of errors, including the lower error bound and the mean.

**Acknowledgements**

We thank the Jungfrau Railway Company for their continuing logistic support during our fieldwork in the central Swiss Alps. We want also to thank the High-Altitude Research Stations Jungfraujoch & Gornergrat for providing us with access to

their research facilities and accommodation. Furthermore, we thank the Swiss National Science Foundation (project No 159299 awarded to F. Schlunegger and A. Ereditato) for their financial support of this research project.

**Author contributions**

AL, FS and AE designed the study

AL developed the code with contributions by MV

AL performed the numerical experiments with support by RN

DM and AL compiled geological data

AA, TA, PS, RN and CP verified the outcome of the numerical experiments

AL wrote the text with contributions from all co-authors

AL designed the figures with contributions by DM

All co-authors contributed to the discussion and finally approved the manuscript

Agostinelli, S., Allison, J., Amako, K., Apostolakis, J., Araujo, H., Arce, P., Asai, M., Axen, D., Banerjee, S., Barrand, G., Behner, F., Bellagamba, L., Boudreau, J., Broglia, L., Brunengo, A., Burkhardt, H., Chauvie, S., Chuma, J., Chytracek, R., Cooperman, G., Cosmo, G., Degtyarenko, P., Dell'Acqua, A., Depaola, G., Dietrich, D., Enami, R., Feliciello, A., Ferguson,
C., Fesefeldt, H., Folger, G., Foppiano, F., Forti, A., Garelli, S., Giani, S., Giannitrapani, R., Gibin, D., Gomez Cadenas, J. J., Gonzalez, I., Gracia Abril, G., Greeniaus, G., Greiner, W., Grichine, V., Grossheim, A., Guatelli, S., Gumplinger, P., Hamatsu, R., Hashimoto, K., Hasui, H., Heikkinen, A., Howard, A., Ivanchenko, V., Johnson, A., Jones, F. W., Kallenbach, J., Kanaya, N., Kawabata, M., Kawabata, Y., Kawaguti, M., Kelner, S., Kent, P., Kimura, A., Kodama, T., Kokoulin, R., Kossov, M., Kurashige, H., Lamanna, E., Lampen, T., Lara, V., Lefebure, V., Lei, F., Liendl, M., Lockman, W., Longo, F.,
Magni, S., Maire, M., Medernach, E., Minamimoto, K., Mora de Freitas, P., Morita, Y., Murakami, K., Nagamatu, M., Nartallo, R., Nieminen, P., Nishimura, T., Ohtsubo, K., Okamura, M., O'Neale, S., Oohata, Y., Paech, K., Perl, J., Pfeiffer, A., Pia, M. G., Ranjard, F., Rybin, A., Sadilov, S., di Salvo, E., Santin, G., Sasaki, T., Savvas, N., et al.: GEANT4 - A simulation toolkit, Nucl. Instruments Methods Phys. Res. Sect. A Accel. Spectrometers, Detect. Assoc. Equip., 506, 250–303, doi:10.1016/S0168-9002(03)01368-8, 2003.

Alvarez, L. W., Anderson, J. A., Bedwei, F. E., Burkhard, J., Fakhry, A., Girgis, A., Goneid, A., Hassan, F., Iverson, D., Lynch, G., Miligy, Z., Moussa, A. H., Sharkawi, M. and Yazolino, L.: Search for Hidden Chambers in the Pyramids, Science (80-. )., 167, 832–839, doi:10.1126/science.167.3919.832, 1970.

Ambrosino, F., Anastasio, A., Bross, A., Béné, S., Boivin, P., Bonechi, L., Cârloganu, C., Ciaranfi, R., Cimmino, L., Combaret, C., D'Alessandro, R., Durand, S., Fehr, F., Français, V., Garufi, F., Gailler, L., Labazuy, P., Laktineh, I., Lénat,
J.-F., Masone, V., Miallier, D., Mirabito, L., Morel, L., Mori, N., Niess, V., Noli, P., Pla-Dalmau, A., Portal, A., Rubinov, P., Saracino, G., Scarlini, E., Strolin, P. and Vulpescu, B.: Joint measurement of the atmospheric muon flux through the Puy de Dôme volcano with plastic scintillators and Resistive Plate Chambers detectors, J. Geophys. Res. Solid Earth, 120, 1–18, doi:10.1002/2015JB011969, 2015.

Ariga, A., Ariga, T., Ereditato, A., Käser, S., Lechmann, A., Mair, D., Nishiyama, R., Pistillo, C., Scampoli, P., Schlunegger,
F. and Vladymyrov, M.: A Nuclear Emulsion Detector for the Muon Radiography of a Glacier Structure, Instruments, 2, 1–13, doi:10.3390/instruments2020007, 2018.

Barnaföldi, G. G., Hamar, G., Melegh, H. G., Oláh, L., Surányi, G. and Varga, D.: Portable cosmic muon telescope for environmental applications, Nucl. Instruments Methods Phys. Res. Sect. A Accel. Spectrometers, Detect. Assoc. Equip., 689, 60–69, doi:10.1016/j.nima.2012.06.015, 2012.

Barrett, P. H., Bollinger, L. M., Cocconi, G., Eisenberg, Y. and Greisen, K.: Interpretation of cosmic-ray measurements far underground, Rev. Mod. Phys., 24, 133–178, doi:10.1103/RevModPhys.24.133, 1952.

Best, M. G.: Igneous and metamorphic petrology, 2nd ed., Blackwell Science Ltd, Malden MA., 2003.

Bezrukov, L. B. and Bugaev, E. V.: Nucleon shadowing effects in photonuclear interaction, Sov. J. Nucl. Phys., 33, 635–641, 1981.

Borchardt-Ott, W.: Kristallographie, Springer Berlin Heidelberg, Berlin, Heidelberg., 2009.

Fano, U.: Penetration of protons, alpha particles, and mesons, Annu. Rev. Nucl. Sci., 13, 1–66, 1963.

Folk, R. L.: Petrology of the sedimentary rocks, Hemphill Publishing Company, Austin TX., 1980.

Fujii, H., Hara, K., Hashimoto, S., Ito, F., Kakuno, H., Kim, S. H., Kochiyama, M., Nagamine, K., Suzuki, A., Takada, Y., Takahashi, Y., Takasaki, F. and Yamashita, S.: Performance of a remotely located muon radiography system to identify the
inner structure of a nuclear plant, Prog. Theor. Exp. Phys., 2013, 073C01, doi:10.1093/ptep/ptt046, 2013.

George, E. P.: Observations of cosmic rays underground and their interpretation, in Process in Cosmic Ray Physics, vol. 1, pp. 395–451., 1952.

George, E. P.: Cosmic rays measure overburden of tunnel, Commonw. Eng., 455, 1955.

Groom, D. E., Mokhov, N. V. and Striganov, S. I.: MUON STOPPING POWER AND RANGE TABLES 10 MeV–100
TeV, At. Data Nucl. Data Tables, 78, 183–356, doi:10.1006/adnd.2001.0861, 2001.

Hayman, P. J., Palmer, N. S. and Wolfendale, A. W.: The Rate of Energy Loss of High-Energy Cosmic Ray Muons, Proc. R.
Soc. A Math. Phys. Eng. Sci., 275, 391–410, doi:10.1098/rspa.1963.0176, 1963.

Hebbeker, T. and Timmermans, C.: A compilation of high energy atmospheric muon data at sea level, Astropart. Phys., 18,
107–127, doi:10.1016/S0927-6505(01)00180-3, 2002.

Higashi, S., Kitamura, T., Miyamoto, S., Mishima, Y., Takahashi, T. and Watase, Y.: Cosmic-ray intensities under sea-water
at depths down to 1400 m, Nuovo Cim. A Ser. 10, 43, 334–346, doi:10.1007/BF02752862, 1966.

Jourde, K., Gibert, D., Marteau, J., de Bremond d'Ars, J., Gardien, S., Girerd, C., Ianigro, J.-C. and Carbone, D.:
Experimental detection of upward going cosmic particles and consequences for correction of density radiography of
volcanoes, Geophys. Res. Lett., 40, 6334–6339, doi:10.1002/2013GL058357, 2013.

Jourde, K., Gibert, D. and Marteau, J.: Improvement of density models of geological structures by fusion of gravity data and
cosmic muon radiographies, Geosci. Instrumentation, Methods Data Syst. Discuss., 5, 83–116, doi:10.5194/gid-5-83-2015,
2015.

Kelner, S. R.: Pair Production in Collisions between Muons and Atomic Electrons, Phys. At. Nucl., 61, 448–456, 1998.

Kelner, S. R., Kokoulin, R. P. and Petrukhin, A. A.: About Cross Section for High-Energy Muon Bremsstrahlung, Prepr.
Moscow Eng. Phys. Inst., 1–32, 1995.

Kelner, S. R., Kokoulin, R. P. and Petrukhin, A. A.: Bremsstrahlung from Muons Scattered by Atomic Electrons, Phys. At.
Nucl., 60, 576–583, 1997.

Kokoulin, R. P. and Petrukhin, A. A.: Analysis of the Cross-Section of Direct Pair Production by Fast Muons, in Proceedings
of the 11th Conference on Cosmic Rays, p. 277, Budapest., 1969.

Kokoulin, R. P. and Petrukhin, A. A.: Influence of the Nuclear Formfactor on the Cross-Section of Electron Pair Production
by High Energy Muons, in Proceedings of the 12th International Conference on Cosmic Rays, pp. 2436–2445, Hobart,
Australia., 1971.

Kusagaya, T. and Tanaka, H. K. M.: Development of the very long-range cosmic-ray muon radiographic imaging technique
to explore the internal structure of an erupting volcano , Shinmoe-dake , Japan, , 215–226, doi:10.5194/gi-4-215-2015, 2015.

Lesparre, N., Gibert, D., Marteau, J., Déclais, Y., Carbone, D. and Galichet, E.: Geophysical muon imaging: feasibility and
limits, Geophys. J. Int., 183, 1348–1361, doi:10.1111/j.1365-246X.2010.04790.x, 2010.

Lesparre, N., Gibert, D., Marteau, J., Komorowski, J.-C., Nicollin, F. and Coutant, O.: Density muon radiography of La
Soufrière of Guadeloupe volcano: comparison with geological, electrical resistivity and gravity data, Geophys. J. Int., 190,
1008–1019, doi:10.1111/j.1365-246X.2012.05546.x, 2012.

Lohmann, W., Kopp, R. and Voss, R.: Energy Loss of Muons in the Range of 1-10000 GeV., 1985.

Mair, D., Lechmann, A., Herwegh, M., Nibourel, L. and Schlunegger, F.: Linking Alpine deformation in the Aar Massif
basement and its cover units – the case of the Jungfrau–Eiger mountains (Central Alps, Switzerland), Solid Earth, 9, 1099–
1122, doi:10.5194/se-9-1099-2018, 2018.

Mando, M. and Ronchi, L.: On the Energy Range Relation for fast Muons in Rock ., Nuovo Cim., 9, 517–529,

doi:10.1007/BF02784646, 1952.

Marteau, J., Carlus, B., Gibert, D., Ianigro, J.-C., Jourde, K., Kergosien, B. and Rolland, P.: Muon tomography applied to active volcanoes, in International Conference on New Photo-detectors, PhotoDet2015, pp. 1–7, Moscow. [online] Available from: http://arxiv.org/abs/1510.05292, 2015.

Matter, A., Homewood, P., Caron, C., Rigassi, D., Stuijvenberg, J., Weidmann, M. and Winkler, W.: Flysch and molasse of western and central Switzerland, in Geology of Switzerland, a guide book: Schweiz Geologica Kommissione, pp. 261–292, Wepf., 1980.

Miyake, S., Narasimham, V. S. and Ramana Murthy, P. V.: Cosmic-ray intensity measurements deep undergound at depths of (800÷8400) m w.e., Nuovo Cim., 32, 1505–1523, doi:10.1007/BF02732788, 1964.

Morishima, K., Kuno, M., Nishio, A., Kitagawa, N., Manabe, Y., Moto, M., Takasaki, F., Fujii, H., Satoh, K., Kodama, H., Hayashi, K., Odaka, S., Procureur, S., Attié, D., Bouteille, S., Calvet, D., Filosa, C., Magnier, P., Mandjavidze, I., Riallot, M., Marini, B., Gable, P., Date, Y., Sugiura, M., Elshayeb, Y., Elnady, T., Ezzy, M., Guerriero, E., Steiger, V., Serikoff, N., Mouret, J.-B., Charlès, B., Helal, H. and Tayoubi, M.: Discovery of a big void in Khufu's Pyramid by observation of cosmic-ray muons, Nature, 552, 386–390, doi:10.1038/nature24647, 2017.

Neddermeyer, S. H. and Anderson, C. D.: Note on the Nature of Cosmic-Ray Particles, Phys. Rev., 51, 884–886, doi:10.1103/PhysRev.51.884, 1937.

Nikishov, A. I.: Energy spectrum of e+e- pairs produced in the collision of a muon with an atom, Sov. J. Nucl. Phys., 27, 677–681, 1978.

Nishiyama, R., Tanaka, Y., Okubo, S., Oshima, H., Tanaka, H. K. M. and Maekawa, T.: Integrated processing of muon radiography and gravity anomaly data toward the realization of high-resolution 3-D density structural analysis of volcanoes: Case study of Showa-Shinzan lava dome, Usu, Japan, J. Geophys. Res. Solid Earth, 119, 699–710, doi:10.1002/2013JB010234, 2014.

Nishiyama, R., Ariga, A., Ariga, T., Käser, S., Lechmann, A., Mair, D., Scampoli, P., Vladymyrov, M., Ereditato, A. and Schlunegger, F.: First measurement of ice-bedrock interface of alpine glaciers by cosmic muon radiography, Geophys. Res. Lett., 44, 6244–6251, doi:10.1002/2017GL073599, 2017.

Oláh, L., Tanaka, H. K. M., Ohminato, T. and Varga, D.: High-definition and low-noise muography of the Sakurajima volcano with gaseous tracking detectors, Sci. Rep., 8, 3207, doi:10.1038/s41598-018-21423-9, 2018.

Olive, K. A.: Review of Particle Physics, Chinese Phys. C, 38, 090001, doi:10.1088/1674-1137/38/9/090001, 2014.

Lo Presti, D., Gallo, G., Bonanno, D. L., Bonanno, G., Bongiovanni, D. G., Carbone, D., Ferlito, C., Immè, J., La Rocca, P., Longhitano, F., Messina, A., Reito, S., Riggi, F., Russo, G. and Zuccarello, L.: The MEV project : Design and testing of a new high-resolution telescope for muography of Etna Volcano, Nucl. Inst. Methods Phys. Res. A, 904, 195–201, doi:10.1016/j.nima.2018.07.048, 2018.

Reyna, D.: A Simple Parameterization of the Cosmic-Ray Muon Momentum Spectra at the Surface as a Function of Zenith Angle, arXiv Prepr. hep-ph/0604145 [online] Available from: http://arxiv.org/abs/hep-ph/0604145, 2006.

Richard-Serre, C.: Evaluation de la perte d'energie unitaire et du parcours pour des muons de 2 à 600 GeV dans un absorbant quelconque., 1971.

Seltzer, S. M. and Berger, M. J.: Evaluation of the collision stopping power of elements and compounds for electrons and positrons, Int. J. Appl. Radiat. Isot., 33, 1189–1218, doi:10.1016/0020-708X(82)90244-7, 1982.

Sternheimer, R. M.: The density effect for the ionization loss in various materials, Phys. Rev., 88, 851–859,

doi:10.1103/PhysRev.88.851, 1952.

Sternheimer, R. M., Seltzer, S. M. and Berger, M. J.: Density effect for the ionization loss of charged particles in various substances, Phys. Rev. B, 26, 6067–6076, doi:10.1103/PhysRevB.26.6067, 1982.

Sternheimer, R. M., Berger, M. J. and Seltzer, S. M.: Density effect for the ionization loss of charged particles in various substances, At. Data Nucl. Data Tables, 30, 261–271, doi:10.1016/0092-640X(84)90002-0, 1984.

Stoer, J. and Bulirsch, R.: Introduction to Numerical Analysis, Springer New York, New York, NY., 2002.

Strunz, H. and Nickel, E.: Strunz Mineralogical Tables. Ninth Edition, Schweizerbart Science Publishers, Stuttgart, Germany., 2001.

Tanaka, H. K. M.: Instant snapshot of the internal structure of Unzen lava dome, Japan with airborne muography, Sci. Rep., 6, 39741, doi:10.1038/srep39741, 2016.

Tanaka, H. K. M., Nagamine, K., Nakamura, S. N. and Ishida, K.: Radiographic measurements of the internal structure of
Mt. West Iwate with near-horizontal cosmic-ray muons and future developments, Nucl. Instruments Methods Phys. Res. Sect. A Accel. Spectrometers, Detect. Assoc. Equip., 555, 164–172, doi:10.1016/j.nima.2005.08.099, 2005.

Tanaka, H. K. M., Kusagaya, T. and Shinohara, H.: Radiographic visualization of magma dynamics in an erupting volcano, Nat. Commun., 5, 1–9, doi:10.1038/ncomms4381, 2014.

Tang, A., Horton-Smith, G., Kudryavtsev, V. A. and Tonazzo, A.: Muon simulations for Super-Kamiokande, KamLAND,
and CHOOZ, Phys. Rev. D - Part. Fields, Gravit. Cosmol., 74, 1–34, doi:10.1103/PhysRevD.74.053007, 2006.

Tioukov, V., De Lellis, G., Strolin, P., Consiglio, L., Sheshukov, A., Orazi, M., Peluso, R., Bozza, C., De Sio, C., Stellacci, S. M., Sirignano, C., D'Ambrosio, N., Miyamoto, S., Nishiyama, R. and Tanaka, H.: Muography with nuclear emulsions - Stromboli and other projects, Ann. Geophys., 60, doi:10.4401/ag-7386, 2017.

Tuttle, O. F. and Bowen, N. L.: Origin of granite in the light of experimental studies in the system NaAlSi3O8-KAlSi3O8-
SiO2-H2O, Geol. Soc. Am. Mem., 74, 1–146, 1958.

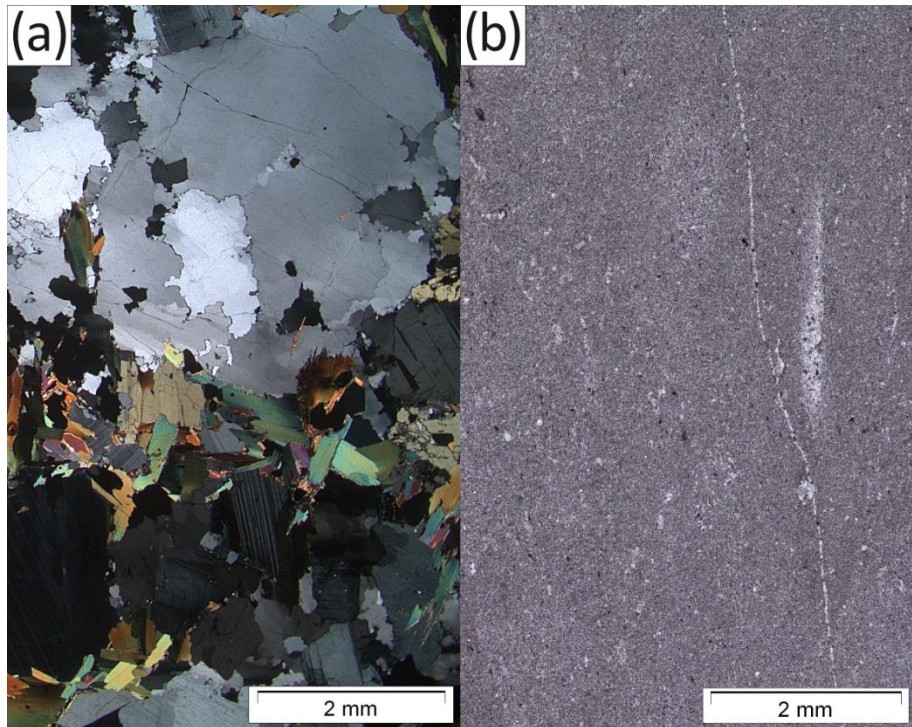

Figure 1: Thin-sections of two representative types of rock in crossed polarised light: (a) Granite, (b) Limestone. The crystal sizes are generally below 4 mm - 5 mm and a few orders of magnitude smaller in the limestone.

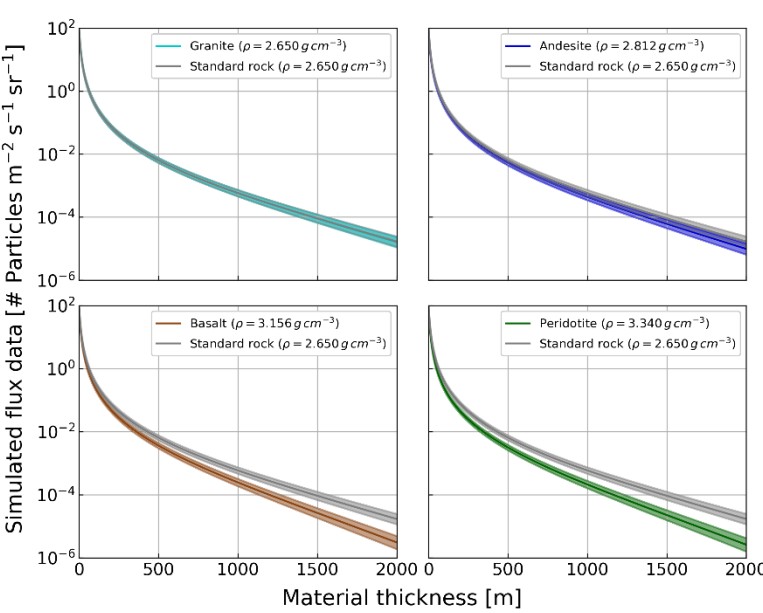


Figure 2: Simulated muon intensity vs. thickness of the four igneous rocks from Table 1 and standard rock. The mean flux is indicated by a bold line and 1 σ bounds are indicated by the shaded area.

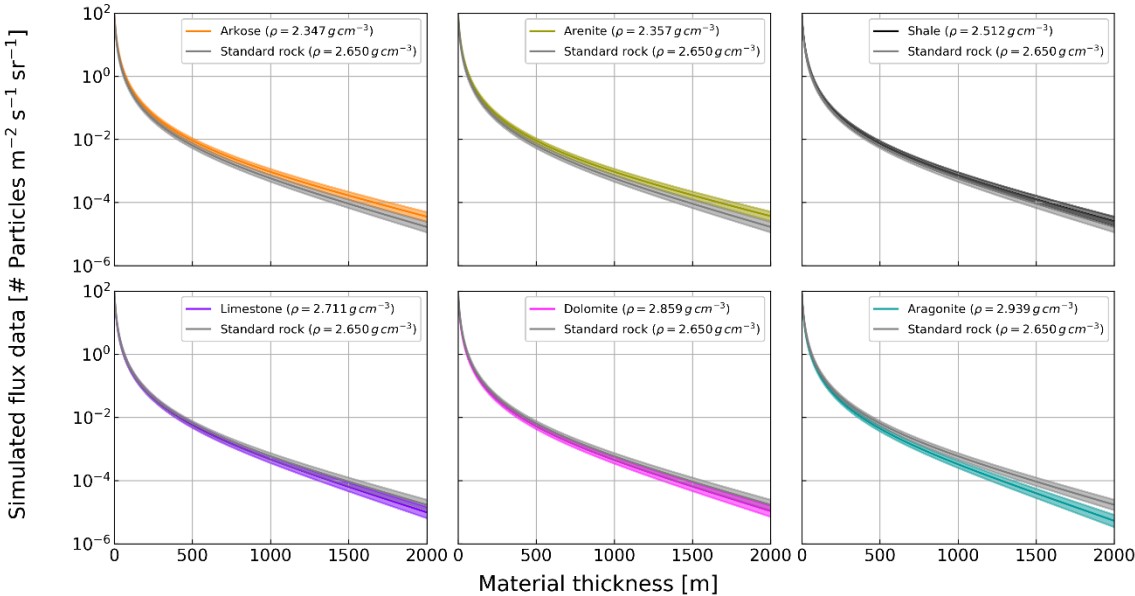

**Figure 3: Simulated muon intensity vs. thickness of the six sedimentary rocks from Table 1 and standard rock. The mean flux is indicated by a bold line and 1 σ bounds are indicated by the shaded area.**

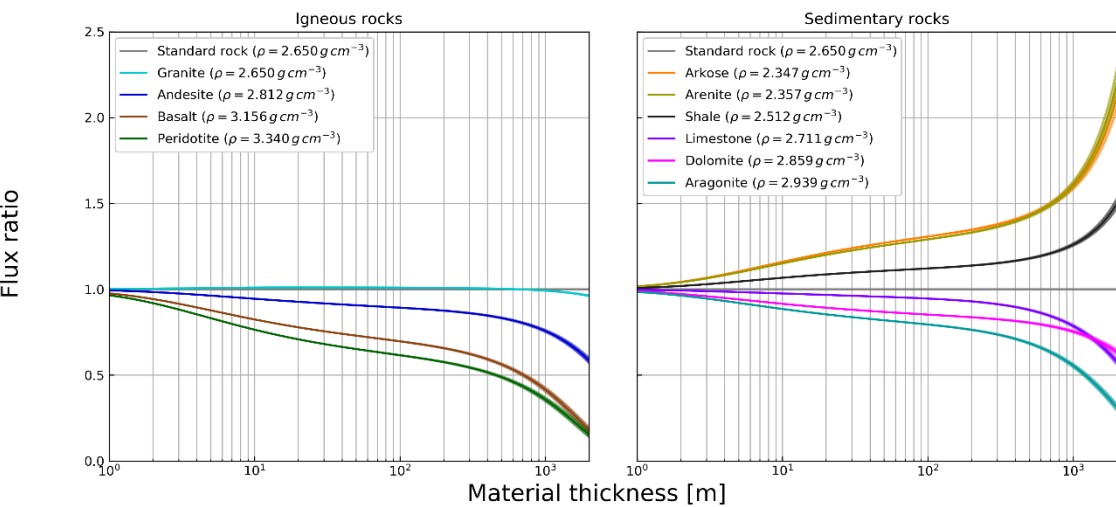

**Figure 4: Ratio of the calculated rock fluxes to a standard rock ($\rho_{SR} = 2.650 \ g \ cm^{-3}$) muon flux for the rocks reported in Table 1 as a function of rock thickness.**

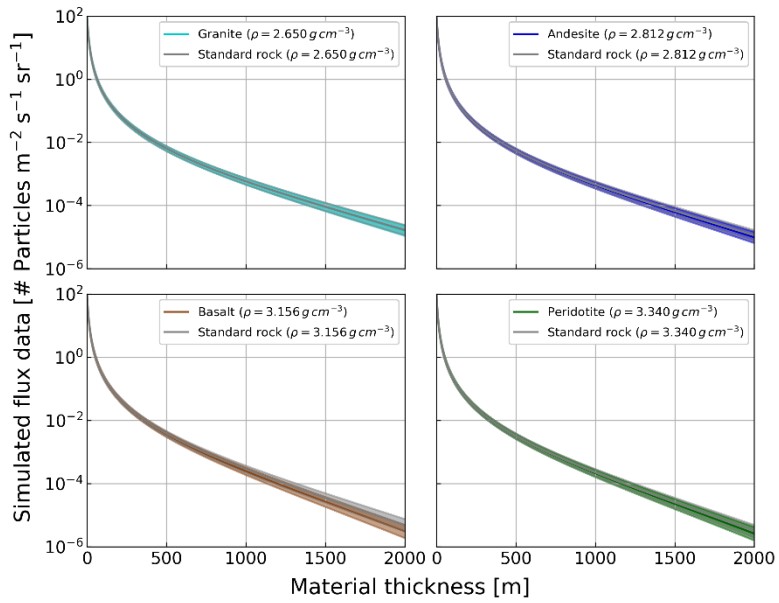


**Figure 5:** Simulated muon intensity vs. thickness of the four igneous rocks from Table 1 and a density modified standard rock. The mean flux is indicated by a bold line and 1 σ bounds are indicated by the shaded area.

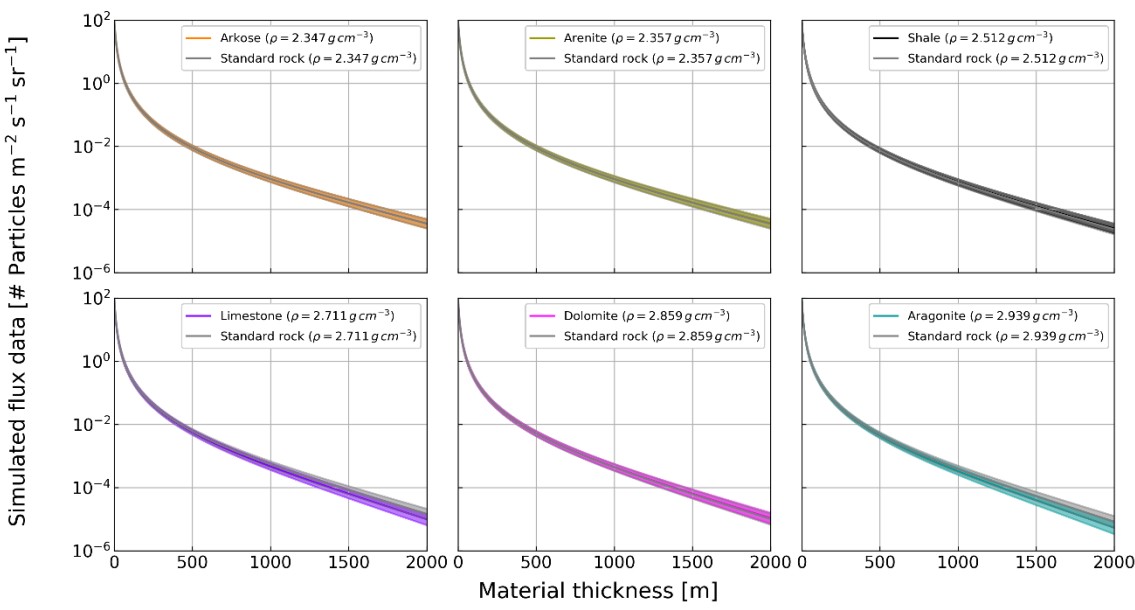

**Figure 6:** Simulated muon intensity vs. thickness of the six sedimentary rocks from Table 1 and a density modified standard rock.
The mean flux is indicated by a bold line and 1 σ bounds are indicated by the shaded area.

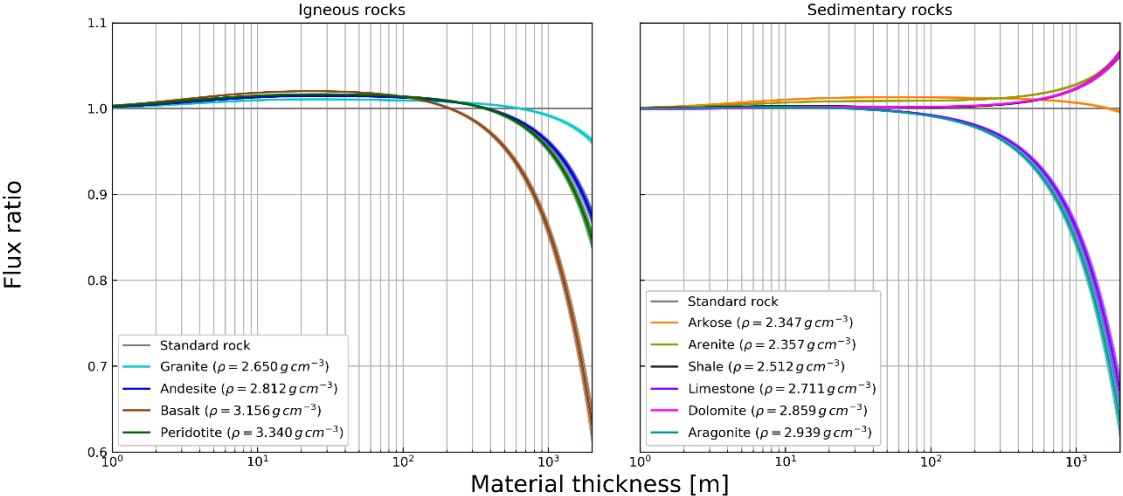

Figure 7: Ratio of the simulated rock fluxes to a standard rock muon flux with the same density as the rock ($\rho_{SR} = \rho_{Rock}$) for all the lithologies in Table 1 as a function of rock thickness.

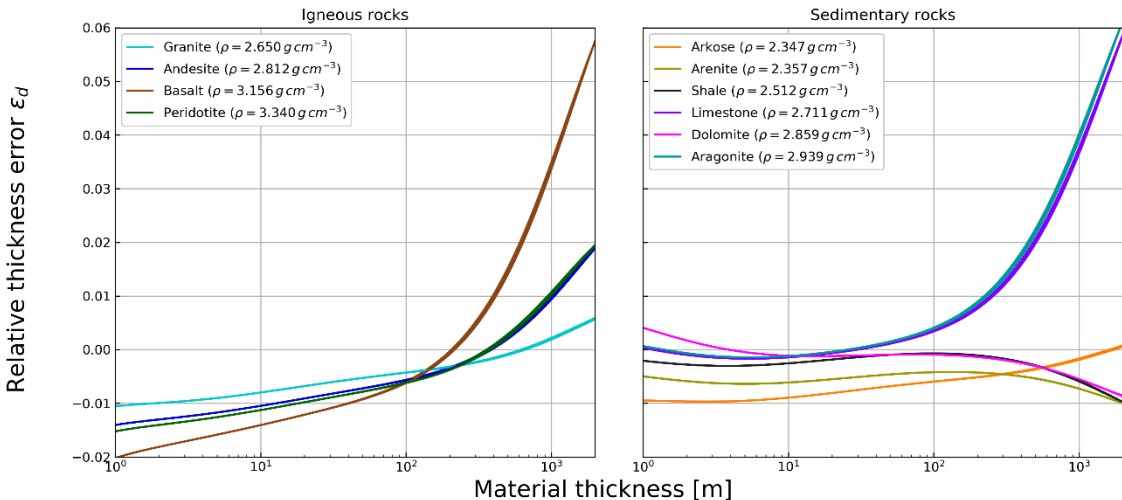

Figure 8: Relative error, which is made in the thickness estimation of a block of rock by assuming a density modified standard rock versus the actual rock thickness.


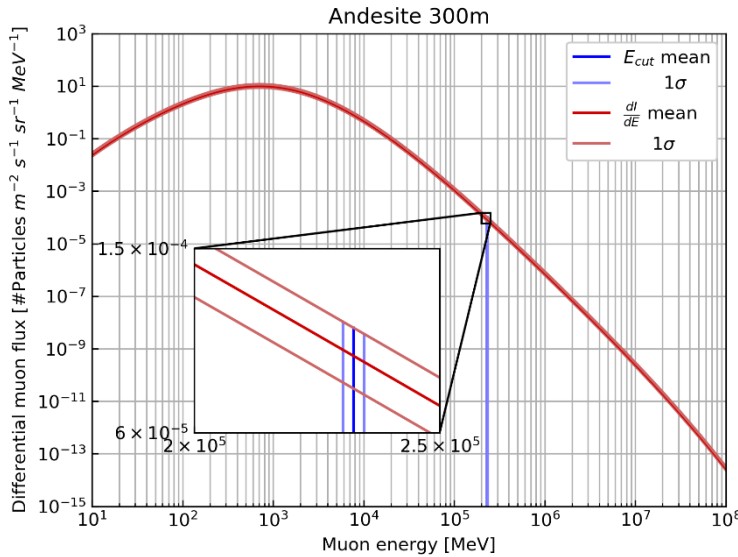

**Figure C1: Differential muon flux as a function of muon kinetic energy. Blue lines indicate the simulated cut-off energy for 300 m of Andesite and its respective propagated error bounds. Red lines show the flux model and its 1 $\sigma$ error bounds.**

**Table 1: Physical parameters of the ten studied rock types and of standard rock.**

| Rock | Density $[g\ cm^{-3}]$ | $\{Z/A\}$ | $\{Z^2/A\}$ | $\{Z^2/A\}/\{Z/A\}$ | $\{I\}$ $[eV]$ |
|---|---|---|---|---|---|
| Standard rock | 2.650 | 0.5000 | 5.500 | 11.0 | 136.40 |
| Igneous rocks | | | | | |
| Granite/Rhyolite | 2.650 | 0.4968 | 5.615 | 11.30 | 145.09 |
| Andesite/Diorite | 2.812 | 0.4960 | 5.803 | 11.70 | 147.77 |
| Gabbro/Basalt | 3.156 | 0.4945 | 6.258 | 12.66 | 154.91 |
| Peridotite | 3.340 | 0.4955 | 5.788 | 11.68 | 149.98 |
| Sedimentary rocks | | | | | |
| Arkose | 2.347 | 0.4980 | 5.563 | 11.17 | 143.73 |
| Arenite (Sandstone) | 2.357 | 0.4993 | 5.392 | 10.80 | 141.04 |
| Shale | 2.512 | 0.4993 | 5.384 | 10.78 | 139.09 |
| Limestone | 2.711 | 0.4996 | 6.275 | 12.56 | 136.40 |
| Dolomite | 2.859 | 0.4989 | 5.423 | 10.87 | 127.65 |
| Aragonite | 2.939 | 0.4996 | 6.275 | 12.56 | 136.40 |

**Table A1: Volumetric percentages of the rock forming minerals within six sedimentary rocks. Qtz: Quartz, Or: Orthoclase, Ab: Albite, An: Anorthite, Cal: Calcite, Dol: Dolomite, Kln: Kaolonite, Mnt: Montmorillonite, Ill: Illite, Clc: Clinochlore**

| Mineral | Arkose | Arenite | Shale | Limestone | Dolomite | Aragonite |
|---|---|---|---|---|---|---|
| Qtz | 56.0 | 89.0 | 17.0 | | | |
| Or | 34.0 | | 2.5 | | | |
| Ab | | | 1.8 | | | |
| An | | | 0.7 | | | |
| Cal | | | | 100.0 | | 100.0 |
| Dol | | | | | 100.0 | |
| Kln | | | 1.7 | | | |
| Mnt | | | 52.7 | | | |
| Ill | | | 22.2 | | | |
| Clc | | | 1.4 | | | |
| Air | 10.0 | 11.0 | | | | |


**Table A2: Volumetric percentages of the rock forming minerals within four igneous rocks. Qtz: Quartz, Or: Orthoclase, Ab: Albite, An: Anorthite, Phl: Phlogopite, Ann: Annite, Mg-Hbl: Magnesium hornblende, Fe-Hbl: Iron hornblende, Aug: Augite, En: Enstatite, Fs: Ferrosilite, Fo: Forsterite, Fa: Fayalite, Jd: Jadeite, Hd: Hedenbergite, Di: Diopside, Spl: Spinel, Hc: Hercynite**

| Mineral | Granite | Andesite | Basalt | Peridotite |
|---------|---------|----------|--------|------------|
| Qtz | 36.1 | 11.7 | | |
| Or | 28.2 | | | |
| Ab | 27.3 | 37.7 | 17.7 | |
| An | | 25.3 | 24.6 | |
| Phl | 2.95 | 4.5 | | |
| Ann | 2.95 | 2.1 | | |
| Mg-Hbl | 2.25 | 4.2 | | |
| Fe-Hbl | 2.25 | 6.4 | | |
| Aug | | 8.1 | 33.8 | |
| En | | | 11.4 | 18.4 |
| Fs | | | 11.1 | 2.0 |
| Fo | | | 0.6 | 60.4 |
| Fa | | | 0.8 | 7.9 |
| Jd | | | | 1.8 |
| Hd | | | | 0.3 |
| Di | | | | 8.0 |
| Spl | | | | 0.9 |
| Hc | | | | 0.3 |