# Peer review of "The effect of rock composition on muon tomography measurements"

_Solid Earth, 2018_

## Referee Comment (RC1) · Anonymous Referee #1 · 7 Jul 2018

This paper studies numerically the sensibility of muon tomography to the type of rock being scanned. The authors calculate the ratio of the flux of muons that should be observed after traversing different types of rock, and compare it to what would be observed if the rock was a "standard type", that is, the reference material used to translate a measurement of muon flux into an average density. To do so, they take into account the mineral composition of each rock type and, for each mineral, the distribution of elements involved, and calculate the total energy loss as a volumetrically averaged energy loss involving each element.

The authors claim that the muon flux measured in common applications are significantly sensitive to the rock type composition and therefore the rock composition should be taken into account for modeling purposes.

[Figure]

The paper is well written and the idea is interesting. The work however lacks some fundamental information necessary to draw the conclusions claimed. There is no discussion regarding the error in the physical models and simulations used to calculate the resulting fluxes, nor how these errors translate into the final parameter studied: the ratio of resulting fluxes between rock types and the standard rock.

Before drawing any conclusions, the authors should provide uncertainty estimations to all the simulations, from the simulation of the incoming flux to the energy loss, and propagate these errors to the final flux ratio.

Furthermore, the methodology developed is limited to a volume averaging of element properties. The results presented involve one sensitivity test that is rather straightforward. Therefore, in its present form the paper does not contain sufficient and sound results: the analysis is limited to one figure where the results lack the uncertainty estimation.

This work would be of a significant impact if the authors could provide, besides the uncertainty in their simulations, real muon measurements associated to different rock types from the field.

Additional comments:

As the authors mention, the incoming flux model is precise only to 10% and this is in the best scenario which corresponds to the vertical direction (zenith angle equal to zero). The question that rises then is, even for the vertical direction used in the paper, what is the purpose of trying to recover an average density in the limit of 2.5 % as the authors use as a threshold.

In a best case scenario where we the incoming flux would be known with a 5 % precision and no errors were associated to the energy loss calculation, one could think that a 5% change in the outgoing flux would be detectable. In this case, only basalt and dolomites would have a significant effect right above the error level, and that only if the

amount of this rock type would be larger than 400 and 500 m of rock respectively.

---

## Referee Comment (RC2) · Anonymous Referee #2 · 6 Sep 2018

The paper is very interesting, well organized and pleasant to read. The authors aim to quantify one of the systematic errors of muography which comes from the chemical composition of rocks. To extract this uncertainty, a novel approach is applied, which is based on the calculation of mineral composition of different rock types and volumetric averaging of energy loss processes. The expected fluxes are calculated based on a muon spectra model and compared to the expected flux after the so-called "standard" rock. The systematic error is found to be less than 2.5 % under the rock thickness of 300 m, where the ionization energy loss process is dominating and it tends to increase with the increase of rock thickness because of the increasing contribution of stochastic energy loss processes.

The scientific comments are summarized in the following points:

[Figure]

1. The main comment is that the systematic uncertainties of the calculations are not presented in the present version of manuscript. This paper presents a novel approach to derive the expected flux and it investigates more rock types to compare with earlier studies, such as [1]. It is suggested to compare the new calculations with another calculations or with GEANT4 simulation. Furthermore, it is suggested to use different flux models, such as [2] or [3], to extract the uncertainty of the calculations.

2. The paper is lack of experimental data taken after rock with known chemical composition. If the authors has any experimental data collected after known rock composition, the calculations should be verified for at least one rock type.

3. The paper could provide more useful information to the muography research community if the authors could extend the study to low-density rocks and soil structures. It is suggested to collect composition information about the different muography targets (underground laboratories, volcanoes) and include them to present study. Furthermore, the extension of the thickness range of flux comparison is suggested up to 3000-3500 meter-standard-rock equivalent.

Further comments are listed here:

Line 33: "recent years this has been done for various volcanoes in Japan (Nishiyama et al., 2014; Tanaka et al., 2005, 2014),": In the recent years the Shinmoe-dake volcano (2015) [4], Unzen lava dome (2016) [5], and most recently (2017-) the Sakurajima volcano [6] have been investigated in Japan. Furthermore, there are ongoing muography experiments at different Italian volcanoes, such as at Etna [7] or Stromboli [8].

Line 38: "500m" -> "500 m" or "500 metres".

Line 105: "I" denotes integrated flux in Eq. (1) and later the mean excitation energy the Appendix B. Maybe it is better to use "F" instead of "I" to denote the integrated flux in Equations (1) and (6).

Line 173: "2.5%" -> "2.5 %" or "2.5\,%"

[Figure]

Line 184: "+/-" -> "$\pm$"

line 210: "600m" -> "600 m" or "600 metres".

Line 226: If my understanding is correct the Avogadro number is used in Eq. A1, however it is defined after Eq. B4.

Line 249: Eq. B2 is suggested to be explained in more details here. All the parameters have to be defined around it.

- The units, e.g. g/cm3 , somewhere written with italic letters, somewhere written with normal letters. It is suggested to unify the written of units.

References:

[1] Juergen Reichbacher and Jeffrey De Jong: Calculation of the Underground Muon Intensity Crouch Curve from a Parameterization of the Flux at Surface, 30TH INTERNATIONAL COSMIC RAY CONFERENCE, ICRC 2007, Merida, Mexico, ftp://ftp.bartol.udel.edu/gaisser/talks/ICRC2007/2.1/icrc0707.pdf

[2] Mengyun Guan et al.: A parametrization of the cosmic-ray muon flux at sea-level, https://arxiv.org/pdf/1509.06176.pdf

[3] Alfred Tang et al.: Muon simulations for Super-Kamiokande, KamLAND, and CHOOZ, Phys. Rev. D 74, 053007, https://journals.aps.org/prd/abstract/10.1103/PhysRevD.74.053007

[4] T. Kusagaya and H. K. M. Tanaka, Development of the very long-range cosmic-ray muon radiographic imaging technique to explore the internal structure of an erupting volcano, Shinmoe-dake, Japan Geosci. Instrum. Method. Data Syst., 4, 215–226, 2015, https://www.geosci-instrum-method-data-syst.net/4/215/2015/

[5] H. K. M. Tanaka: Instant snapshot of the internal structure of Unzen lava dome, Japan with airborne muography, Scientific Reports 6:39741 DOI:10.1038/srep39741

[6] L. Oláh et al.: High-definition and low-noise muography of the Sakurajima volcano with gaseous tracking detectors http://www.nature.com/articles/s41598-018-21423-9

[7] D. Lo Presti et al.: The MEV project: design and testing of a new high-resolution telescope for Muography of Etna Volcano, https://arxiv.org/pdf/1805.11612.pdf

[8] Valeri Tioukov et al.: Muography with nuclear emulsions - Stromboli and other projects, ANNALS OF GEOPHYSICS, 60, 1, 2017, S0111; doi:10.4401/ag-7386

---

## Editor Comment (EC1) · M. J. Heap (Editor) · 6 Sep 2018

Dear authors,

As you can see, I have now received two reviews of your manuscript. In general, they are positive about the goal of the manuscript. Reviewer #1 describes the paper as "an interesting idea" and reviewer #2 states that the paper is "very interesting". Nevertheless, they both raise important points that should be rectified before I consider your manuscript suitable for publication in Solid Earth. In particular, both reviewers highlight the lack of an error/uncertainty analysis as one of the main drawbacks of the submitted manuscript. Both reviewers also mention that muon measurements for different rock types of known composition would also greatly boost the impact of this study. I consider

it important that these two points are addressed in your revised manuscript. If you are willing, please now prepare a point-by-point rebuttal letter and a revised manuscript.

I thank you again for considering Solid Earth as a platform for your work.

Mike Heap (Topical Editor of Solid Earth)

---

## Author Comment (AC1) · 3 Oct 2018

We thank the anonymous reviewer for the comments and the positive reception of our idea.

In our view, reviewer 1 raises 4 major points. The first one concerns the propagation of any uncertainty in the energy loss calculations and the flux model to the simulated data and a subsequent error propagation of these calculated fluxes to the flux ratio. We completely agree with the reviewer that the related uncertainties exist and should be propagated to the simulated flux. This can be seen in Eq. (1), where the lower integration boundary, i.e. $E\_cut$, carries the error of the interaction cross-sections. Furthermore, the integrand, i.e. $dI/dE$, has also an error attributed to it. As a consequence,

the simulated flux, i.e. the left side of Eq. (1), has to be represented by a probability distribution. If the errors in the nominator and the denominator were independent from each other, we would have to fully propagate the error to flux ratio according to the gaussian law of uncertainty propagation. However, as both simulations were obtained with the exact same energy-loss and flux models, the errors are even correlated one to one. Thus, the error propagation can be treated much more efficiently. We will visualise these points by adding an additional figure. We will then show how these errors are propagated to our final parameter, which is the flux ratio. We will address this issue in a new section, which is dedicated to the discussion of the involved errors in our study. Following the reviewer's advice, we will also add errors to our existing flux ratio figures.

A further point mentions the limitation of our approach to a volume averaging of element properties. We apologise if we might not have completely and clearly conveyed the idea of our method. We would like to clarify that our rock model incorporates two different kinds of averaging. Element properties are averaged by their mass to crystal properties. Only then did we employ a volume averaging of these crystal properties to rock properties. That being said, we have to add that the volume averaging is a procedure, which is equivalent to the mass averaging. In fact, we derived the volume averaging approach through a mass averaging of crystal properties as can be seen from Eq. (B16) onwards. We justify this approach through our observations, where a rock is made up of several minerals that have their own spatial extent. In turn, each mineral has its own elemental composition and crystallographic structure. This approach allows for much more detailed models to be used for the description of rocks, than those that are based on bulk compositions which are only represented by oxide fractions. We suggest that this different view better serves the need in muon tomographic studies. We will thus adapt our manuscript accordingly to express the idea of our approach more clearly. We will gladly provide also the energy loss equations for a rock that has been obtained by a mass averaging procedure of crystal properties in the supplementary material.

The third issue states that experimental data could greatly improve the impact of our study. We agree with the fact that our work would benefit if experimental data would confirm our theoretical findings. However, we acknowledge that we cannot offer quantitative measurements to test and constrain our model, mainly because the required data is not available. We do see major advances in this field if quantitative data on the dependency of muon flux attenuation on lithology would be available. Nevertheless, we based our inferences on the same conceptual framework as those that have been applied for any other material, including standard rock. As a result of this, we find differences if the rock parameters are changed. Therefore, we see the need that interpretations of muon fluxes need to be adjusted according to the geological architecture. We implement this in our work by discussing the role of experimental data for this sensitivity study.

The last matter raised by the reviewer is the significance of our findings in light of the prevalence of the flux model error. We agree with the reviewer that in a standard, present-day muon tomographic experimental setup (i.e. measuring the muon flux on the "back"-side of the target and assuming a flux model for the muon flux in "front" of the target), the dominating systematic error originates generally from the flux model, such that the compositional error would be regarded as negligible. However, one could imagine an experimental setup, where the flux in front of the target is also measured. Thus, the necessity of imposing a flux model disappears and one is limited only by the measurement accuracy. It is also possible that the community measuring the muon flux will improve their models in a way that the errors associated with it become smaller. An extreme view would even be to draw inferences on rock composition from muon flux measurements. Either way, the problem of the sensitivity to compositional changes will resurface in the future as this technology will be constantly refined. We address this issue more carefully in our work, such that we can emphasise that our technical developments were not tied to a rigid experimental setup and might be useful for future studies.

---

## Author Comment (AC2) · 3 Oct 2018

We thank the anonymous referee for the constructive, insightful and detailed comments that we think are very helpful to improve the manuscript. In this Author Comment we will first address the major points that were raised, alongside our answer and how we improve the manuscript by implementing the reviewer's suggestions.

Reviewer 2 raises 3 key issues, the first of which being the uncertainties of the simulation which propagate to the flux ratio, i.e. Eq. (6). This point can be split into two separate problems. On one hand the propagation of the systematic error of the used flux model and on the other hand the propagation of the systematic errors that are present in the cross-sections used for energy loss calculations. Furthermore, it is sug-

gested to use another flux model to validate our results. We agree with the reviewer that, as can be seen in Eq. (1), there are inherent model error in both the lower integration boundary, i.e. E_cut, that stems from errors in the interaction cross-sections, as well as in the integrand, i.e. dI/dE , representing the error on the cosmic-ray flux model. Consequently, the simulated flux, i.e. left side of Eq. (1) must be represented by a probability distribution. However, we want to emphasise the fact, that the flux ratio is a fraction of two simulations, in which the same flux model and the same interaction cross-sections have been used. The only model parameters that have been changed were the density and the compositional parameters Z & A of the material (atomic number & atomic weight). This implies that the errors on the flux simulation in the numerator and in the denominator are correlated one to one. Furthermore, the errors within the simulation can be assumed to be of gaussian nature and independent from each other. These prerequisites allow us to treat the error on the flux ratio in an efficient way. In the revised manuscript, we will prepare an additional figure that will visualise the above reasoning and clarify the origin and the propagation of these errors to the flux ratio. This will be implemented in a new section dedicated to the discussion of the role of uncertainty in our study. Moreover, we will modify our flux ratio figures in such a way that the error will be clearly displayed. We also follow the reviewer's suggestion and show the same calculations for another flux model.

A second issue concerns the lack of experimental data within our study and the request to test our sensitivity study against it. We acknowledge that we cannot offer quantitative measurement results to test and constrain our model with experimental data, mainly because the required data is not available. We do see major advances in this field if quantitative data on the dependency of muon flux attenuation on lithology would be available. Nevertheless, we based our inferences on the same conceptual framework as those that have been applied for standard rock, and as a result of this, we find differences if the rock variables are changed. Therefore, we see the need that interpretations of muon fluxes need to be adjusted according to the geological architecture. We implement this in our revised manuscript by discussing the role of experimental

data for this sensitivity study.

The third point addresses the amount of studied rock types, as to the community it could be more useful to have even more different rocks and soil structures covered by this study. Beyond that, it is requested to extend the rock thickness to 3000-3500 metre-standard-rock equivalent. (which corresponds roughly to 1100-1300 metres of rock). The rock compositions in our study were chosen such that they cover the most important varieties of igneous and sedimentary rocks. All other rocks, with the exception of metamorphic rocks, have a composition somewhere intermediate between these corner stones. The fact of a rock exhibiting a notable lower density can predominantly be traced back to its porosity, i.e. the inclusion of air (or in the case of volcanoes also carbon dioxide) between the minerals. As the material density is very susceptible to changes in porosity, the compositional parameters $Z/A$ and $Z^2/A$ are not affected by much. This can be understood from Eq. B23 and Eq. B24, where the density of the j-th mixture component acts as a weight to the total compositional parameters. This means that the (admittedly rather low $Z^2/A$ value of air and CO2, both are around 3.7, compared to 5-6 for rocks) has practically no effect as it becomes downweighed by a factor $\sim$1000 (fraction between rock and air density). It follows that the low-density rock variants, after a density normalisation, are almost equivalent to their pure counterparts. We want to stress, that a similar problem exists already in the manuscript, as can be seen in Fig. 4. Limestone and Aragonite, that show a density difference of $\sim$10% have the exact same composition. After a density normalisation the difference at 1km thickness is below 1%. We gladly accept the suggestion to extend the simulated rock thicknesses. It remains however questionable if it makes sense graphically, as the figures in the manuscript use a log scale for the thickness, such that 200m more would not be a large extension. On the other side, an extension by a whole log-step, i.e. to 10km, although scientifically interesting, would be practically irrelevant, as the resulting muon flux would be too low for any practical purpose. We add a more detailed discussion on how to cope with low density rocks in the manuscript, as the method remains valid. As for the extension of the simulated rock thickness, we ought to seek a

compromise between scientific interest, practical relevance and figure readability.

We appreciate all additional suggestions by the reviewer and we will adapt our revised manuscript accordingly.

---

## Author Response (AR2)

We thank the reviewers for the constructive and insightful comments as well as for the positive reception of our idea. We considered the reviews helpful in improving the clarity of our manuscript's message and the quality of the underlying science, and in increasing the robustness of our conclusions. In the following, we provide individual responses to the reviewer's concerns and start with a general outline of how we have updated and improved our manuscript, which is followed by detailed comments on specific issues, raised by the reviewers. Please note that the reviewer comments are indicated by italics. Additionally, we made a few minor changes to correct for some small errors that were not pointed out by the reviewers. At the end, we provide a track-changed version of our manuscript.

**General response to Reviewers 1 & 2**

We thank the anonymous referees for the constructive comments that helped to strengthen our final conclusions. We implemented these changes according to the suggestions of the reviewer. Particularly, both reviewers raise two similar points:

First, the reviewers are concerned about the lack of an error propagation of the uncertainties, which are present in the interaction cross-sections as well as in the flux model, to the flux ratio. Reviewer 1 states: *"[…] there is no discussion regarding the error in the physical models and simulations […]"* and *"Before drawing any conclusions, the authors should provide uncertainty estimations to all the simulations […] and propagate these errors to the final flux ratio."* Reviewer 2 states: *"The main comment is that the systematic uncertainties of the calculations are not presented […]"*

We completely agree with the reviewer that the related uncertainties exist and should be propagated to the simulated flux and finally also to the flux ratio.

We thus have reworked all affected figures (i.e. Fig. 2 onwards) to include an estimation of the error that derives from the model-inherent errors (i.e. flux model and interaction cross-sections). Moreover, the systematic errors are now explicitly stated in the manuscript (see Sect. 2.2 and Sect. 2.3) and the discussion treats the effect of the errors. We also added a Section in the Appendix (see Appendix C), to show our rationale behind the error propagation.

Secondly, both reviewers mention the lack of experimental data. Reviewer 1 states: *"This work would be of significant impact if the authors could provide […] real muon measurements associated to different rock types from the field."* Reviewer 2 states: "The paper is lack of experimental data […]" and "If the authors has any experimental data […] the calculations should be verified […]"

We agree with the reviewers on the fact that our work would benefit if experimental data would confirm our theoretical findings. However, we acknowledge that we cannot offer quantitative measurements to test and constrain our model, mainly because the required data is not available. Nevertheless, our inferences are based on

the same theoretical framework that has already been used for other materials, including standard rock. Thus, we still gain insight into this problem, even without experimental data at hand.

We address this point by adding a dedicated paragraph in our discussion section.

A third point mentions the limitation of our approach to a volume averaging of element properties. Reviewer 1 states: *"[…], the methodology developed is limited to a volume averaging of element properties."*

We apologise if we might not have completely and clearly conveyed the idea of our method. We would like to clarify that our rock model incorporates two different kinds of averaging. Element properties are averaged by their mass to crystal properties. Only then did we employ a volume averaging of these crystal properties to rock properties. That being said, we have to add that the volume averaging is a procedure, which is equivalent to the mass averaging. In fact, we derived the volume averaging approach through a mass averaging of crystal properties as can be seen from Eq. (B16) onwards. We justify this approach through our observations, where a rock is made up of several minerals that have their own spatial extent. In turn, each mineral has its own elemental composition and crystallographic structure. This approach allows for much more detailed models to be used for the description of rocks, than those that are based on bulk compositions which are only represented by oxide fractions. We suggest that this different view better serves the need in muon tomographic studies.

We added clarifications in the main body of the manuscript, as well as at the end of Appendix B, to stress that a volumetric averaging and a mass averaging are equivalent procedures. The additional equations are presented in the supplementary material.

A further issue concerns the significance of our findings in light of the prevalence of the flux model error. Reviewer 1 states: *"[…] the incoming flux model is precise only to 10% […]"* and *"[…] what is the purpose of trying to recover an average density […] of 2.5% […]"*

We agree with the reviewer that in a standard, present-day muon tomographic experimental setup (i.e. measuring the muon flux on the "back"-side of the target and assuming a flux model for the muon flux in "front" of the target), the dominating systematic error originates generally from the flux model, such that the compositional error would be regarded as negligible. However, one could imagine an experimental setup, where the flux in front of the target is also measured. Thus, the necessity of imposing a flux model disappears and one is limited only by the measurement accuracy. It is also possible that the community measuring the muon flux will improve their models in a way that the errors associated with it become smaller. An extreme view would even be to draw inferences on rock composition from muon flux measurements. Either way, the basic problem, i.e. the sensitivity of the measured muon flux to compositional changes, persists in every application of this technology and it is only a question of the experimental/study design if this effect appears. Thus, this issue will resurface in the future as this technology will be constantly refined.

As an example of how our calculations can be applied to an existing muon tomography experiment, we added a new figure (Fig. 8) that shows how large the error on an estimated thickness would be, by assuming a density modified standard rock instead of its realistic counterpart.

Lastly, reviewer 2 suggests extending the calculations to greater thicknesses as he states: *"[…] the extension of the thickness range of flux comparison is suggested up to 3000-3500 meter-standard-rock equivalent."*

We gladly follow this suggestion and extend our calculations up to 2 km of thickness, as 3500 metre-standard-rock-equivalent roughly correspond to 1.5 km of arkose (the lowest density rock in our calculations) or 1 km of peridotite (our highest density rock). Thus, the upper boundary of 2 km encompasses the suggested thicknesses for all rocks.

**Line by Line responses to reviewer 2:**

Line 33: *"[…] In the recent years the Sinmoe-dake volcano […], Unzen lava dome […] and most recently […] the Sakurajima volcano […] have been investigated in Japan. Furthermore, there are ongoing muography experiments at different Italian volcanoes, such as Etna […] or Stromboli […]."*

Response: We gladly incorporate these other studies in our introduction.

Line 38: *"500m" -> "500 m" or "500 metres"*

Response: Changed "500m" to "500 m".

Line 105: *" "I" denotes integrated flux in Eq. (1) and later the mean excitation energy the Appendix B. Maybe it is better to use "F" instead of "I" to denote the integrated flux in Equations (1) and (6)."*

Response: We thank the reviewer for this clarifying comment and gladly adapt our notation.

Line 173: *"2.5%" -> "2.5 %" or "2.5\,%"*

Response: Changed "2.5%" to "2.5 %" everywhere in the manuscript as well.

Line 184: *"+/- " -> "$\pm$"*

Response: Changed "+/-" to its equation alternative "$\pm$"

Line 210: *"600m" -> "600 m" or "600 metres"*

Response: Changed "600m" to "600 m".

Line 249: *"Eq. B2 is suggested to be explained in more details here. All the parameters have to be defined around it."*

Response: We added a few lines explaining the rest of the parameters in equation B2. However, we think that a detailed explanation is outside the scope of our paper and the reader is referred to dedicated literature.

**Additional changes by authors:**

In addition to the above changes we corrected some inconsistencies, which we came across while revising our manuscript.

First, we realised, that the flux values of the carbonates (i.e. Dolomite, Aragonite and Limestone) were generally too high. We found the reason for this to be the missing element Carbon in our calculations. This error has been remedied and the concerning figures have been altered. The general effect of this change was a shift of the affected rocks towards lower flux ratios. Dolomite falls now in line with Shale and Arenite and Limestone/Aragonite behave more similar to basalt, as their general $\{Z^2/A\}$ – value suggests. We changed also the example, which before was dolomite, to limestone to exemplify the effect of a worst case of a compositional error.

We corrected also the values in Table 1. These errors were due to a propagation of rounding errors. However, the magnitude of the corrections is mainly below 1 % of the nominal value and does thus not affect our final conclusions.

[revised manuscript text omitted]